



# Correction of a lunar irradiance model for aerosol optical depth retrieval and comparison with star photometer

Roberto Román[1], Ramiro González[1], Carlos Toledano[1], África Barreto[2,3,1], Daniel Pérez-Ramírez[4,5], Jose A. Benavent-Oltra[4,5], Francisco J. Olmo[4,5], Victoria E. Cachorro[1], Lucas Alados-Arboledas[4,5], and Ángel M. de Frutos[1]

[1]Group of Atmospheric Optics (GOA-UVa), Universidad de Valladolid, 47011, Valladolid, Spain
[2]Izaña Atmospheric Research Center, Meteorological State Agency of Spain (AEMET), Izaña, Spain
[3]Cimel Electronique, Paris, France
[4]Department of Applied Physics, Universidad de Granada, 18071, Granada, Spain
[5]Andalusian Institute for Earth System Research, IISTA-CEAMA, Granada, Spain

**Correspondence:** Roberto Román (robertor@goa.uva.es)

**Abstract.** The emergence of Moon photometers is allowing measurements of lunar irradiance over the world and increasing the potential to derive aerosol optical depth (AOD) at night-time, that is very relevant in polar areas. Actually, new photometers implement the latest technological advances that permit lunar irradiance measurements together with classical Sun photometry measurements. However, a proper use of these instruments for AOD retrieval requires accurate time-dependent knowledge of the extraterrestrial lunar irradiance over time, due to its fast change throughout the Moon's cycle. This paper uses the RIMO model (an implementation of the ROLO model) to estimate the AOD at night-time assuming that the calibration of the solar channels can be transferred to the Moon by a vicarious method. However, the obtained AOD values using a Cimel CE318-T Sun/sky/Moon photometer for 98 pristine nights with low and stable AOD at the Izaña Observatory (Tenerife, Spain) are not in agreement with the expected (low and stable) AOD values, estimated by linear interpolations from daytime values obtained during the previous evening and the following morning. Actually, AOD calculated using RIMO shows negative values and with a marked cycle dependent on the optical airmass. The differences between the AOD obtained using RIMO and the expected values are assumed to be associated with inaccuracies in the RIMO model, and these differences are used to calculate the RIMO correction factor (RCF). The RCF is a proposed correction factor that, multiplied by RIMO value, gives an effective extraterrestrial lunar irradiance that provides the expected AOD values. The RCF varies with the Moon phase angle (MPA) and with wavelength, ranging from 1.01 to 1.14, which reveals an overall underestimation of RIMO to the lunar irradiance. These obtained RCF values are modeled for each photometer wavelength to a second order polynomial as function of MPA. The AOD derived by this proposed method is compared with the independent AOD measurements obtained by a star photometer at Granada (Spain) for two years. The mean of the Moon-star AOD differences are between -0.015 and -0.005 and the standard deviation between 0.03 and 0.04 (which is reduced to about 0.01 if one month of data affected by instrumental issues is not included in the analysis), for 440, 500, 675 ad 870 nm; however, for 380 nm, the mean and standard deviation of these differences are higher. The Moon-star AOD differences are also analyzed as a function of MPA, showing no significant dependence.



# 1 Introduction

Atmospheric aerosols interact with radiation by scattering and absorption mechanisms and with clouds mainly by acting as
cloud condensation nuclei, which modify the cloud properties like cloud lifetime or droplet size (Boucher et al., 2013). These
issues make the aerosol direct and indirect effects to play a crucial role in the Earth's energy budget, being its impact still subject
to large uncertainties (IPCC, 2014) due to the large aerosol diversity in size, chemical composition or spatial distribution.
These current uncertainties in climate models point out the need to monitor aerosol properties and motivate the study of their
interaction mechanisms with the Earth-Atmosphere system (Myhre et al., 2013). In addition, the impact of aerosols is important
in several fields such as: air quality and human health (Davidson et al., 2005); marine and land ecosystems (Koren et al., 2006;
Ravelo-Pérez et al., 2016); primary productivity (Jickells et al., 2005), precipitation (Twomey, 1977; Stevens and Feingold,
2009); solar energy production (Neher et al., 2017); or air traffic (Flentje et al., 2010), among others.

Most of the aerosol studies focused on the aerosol role in the climate change field are based on daytime measurements.
However, the knowledge of aerosol properties at night-time is also important, especially in polar areas, where a lack of aerosol
observations over winter still exists (Herber et al., 2002; Mazzola et al., 2012; Graßl and Ritter, 2019). In addition, a large
fraction of aerosols at night-time remains in the residual layer, which may even act as a source for aerosol formation into the
boundary layer the next day (Sun et al., 2013; Liu et al., 2020). Moreover, the lack of ultraviolet (UV) radiation at night-time
should reduce the events of new particle formation at night since it is reported that solar UV radiation helps to induce some
nucleation events (Petäjä et al., 2009). Finally, the direct effect of aerosols on solar radiation at night-time is avoided, but the
aerosols presented at night-time can profoundly modify the longwave balance by means of the change in cloud properties and
the impact on the longwave radiation absorbed by clouds, which is back-emitted to the Earth's surface.

Two of the most important/used aerosol properties in climate-change studies and modelling are: the aerosol optical depth
(AOD), which represents the light extinction in the atmospheric column caused by aerosols; and the so called Ångström
Exponent (AE; Angström 1961) which quantifies the AOD spectral variation. These two parameters provide information about
the aerosol load and the particle size predominance, respectively. Moreover, AOD values are useful to estimate other aerosol
properties in combination with other measurements (e.g., sky radiance and lidar signal) or even without them (Lopatin et al.,
2013; Torres et al., 2017; Román et al., 2017, 2018; Benavent-Oltra et al., 2019). However, ground-based AOD values are
usually obtained by solar radiation extinction measurements. In-situ instrumentation is useful to obtain aerosol properties
at night-time but they are usually representative only of the aerosol at ground level, with the exception of airborne in-situ
measurements (e.g., Remer et al., 1997). Some remote sensing techniques used to derive the aerosol properties at night-time
are the Raman lidar systems (Ansmann et al., 1990), which provide AOD but also vertically-resolved extinction profiles; and
the star photometers, which derive the AOD from star light extinction measurements (Pérez-Ramírez et al., 2008a; Baibakov
et al., 2015). The availability of star photometers is very scarce, existing approximately only five star photometers at present in
the world operating for aerosol monitoring (Barreto et al., 2019). Recent technical advances allow accurate measurements of
direct lunar irradiance (Berkoff et al., 2011; Barreto et al., 2013), therefore the emerging Moon photometry technique appears





as a plausible and operative alternative for AOD calculation at night-time. One disadvantage of Moon photometry is that lunar irradiance is only recorded from first to third Moon quarter, which implies a lack of data during half of the Moon cycle.

Some Moon photometers are capable to take measurements of solar and lunar direct irradiances, like the CE318-T Sun/sky/Moon photometer (*Cimel Electronique S.A.S.*), which is the standard instrument in AERONET (AErosol RObotic NETwork; Holben et al., 1998). This fact allows the well established calibration of the solar channels in the AERONET protocols to be transferred to the Moon (Barreto et al., 2016; Li et al., 2016). The main difference between Sun and Moon photometry is that the extraterrestrial lunar irradiance quickly varies even in the course of one night while the extraterrestrial solar irradiance is more stable, with a smooth variation over the year. This remarks the need for knowledge of accurate extraterrestrial lunar irradiance values and their temporal variations. To this end, some models are used, being ROLO (RObotic Lunar Observatory, Kieffer and Stone (2005)) the most widely used in the literature. Here we make use of one implementation of the ROLO named as RIMO (ROLO Implementation for Moon's Observation; Barreto et al., 2019). The irradiance from these models is usually assumed as true for the AOD calculation, however different authors reported some uncertainties and biases in these models (e.g.,Viticchie et al. 2013; Lacherade et al. 2014; Barreto et al. 2017; Geogdzhayev and Marshak 2018).

In this framework, the main objective of this work is to evaluate the RIMO accuracy from the differences between the expected AOD in a pristine environment (where AOD is assumed to be low and stable) and the AOD derived by the RIMO with the CE318-T in the same place. The purpose behind this evaluation is to find a correction of the RIMO model that provides an effective lunar extraterrestrial irradiance, which will be assumed as true, useful at least to derive accurate AOD values in the CE318-T bands using the operative Sun-Moon calibration transfer technique. In addition, this paper aims at studying the performance of the AOD obtained with Moon photometry using the proposed RIMO correction, through a comparison with the AOD from a star photometer.

This paper is structured as follows: Section 2 introduces the sites and instrumentation used in this paper; Section 3 presents the development of the proposed correction on the RIMO lunar irradiance model, while the comparison of the AOD derived using this correction and the one obtained by a star photometer is shown in Section 4. Finally, Section 5 summarizes the main conclusions of this work.

## 2 Sites, Instrumentation and data

### 2.1 Sites

The RIMO correction proposed in this paper is based on photometer data recorded at the Izaña Meteorological Observatory (IZO; 28.309ºN; 16.499ºW; 2401 m a.s.l.) in the Canary Islands (Tenerife, Spain), which is managed by the Spanish Meteorological Agency (AEMET). This high-mountain observatory is representative most of the time of the subtropical free troposphere over the North Atlantic, because of its location in the descending branch of the Hadley's cell (Rodríguez et al., 2009; Cuevas et al., 2019). Pristine skies, dry atmospheric conditions and atmospheric stability prevail throughout the year, as a consequence of the quasi-permanent temperature inversion layer, normally located below the Izaña's level. This situation prevents the vertical transport of anthropogenic pollution from lower levels (Rodríguez et al., 2009).





In terms of AOD, pristine conditions are prevalent in this station, with AOD at 500 nm below 0.1 and AE above 0.6 (Guirado-
Fuentes, 2015). Relatively high AOD conditions due to the Saharan dust transport from North Africa sources to the Atlantic
Ocean above the trade wind inversion are prevalent in summer (Basart et al., 2009; Rodríguez et al., 2011), associated typically
with the presence of coarse particles (AE below 0.25) and AOD at 500 nm above 0.1 (Basart et al., 2009; García et al., 2012;
Guirado-Fuentes, 2015). These privileged conditions make Izaña Observatory a suitable place for calibration and validation
activities (Toledano et al., 2018). Notwithstanding, Izaña is a calibration site for the GAW-PFR (Global Atmosphere Watch
precision-filter radiometer) and AERONET networks (Cuevas et al., 2019), holding a comprehensive measurement programme
for atmospheric composition monitoring, being designated by the WMO (World Meteorological Organisation) as a CIMO
(Commission for Instruments and Methods of Observation) testbed for aerosols and water vapour remote sensing instruments
(WMO, 2014). More details about monitoring programmes at Izaña can be found in Cuevas et al. (2017).

The star photometer measurements of this paper were carried out at the University of Granada (UGR) experimental station,
which is the main station of the three belonging to AGORA (Andalusian Global ObseRvatory of the Atmosphere). This station
is located at the Andalusian Institute for Earth System Research/IISTA-CEAMA (37.164°N; 3.605°W; 680 m a.s.l.). The
UGR station operates many remote sensing instruments in the framework of the ACTRIS (Aerosols, Clouds and Trace Gases,
www.actris.eu) infrastructure, being the star photometry data at UGR the only available of this type in ACTRIS. The UGR
experimental site is located in city of Granada (Spain), which is a medium-size city (535000 inhabitants in all metropolitan
area) at south-eastern Spain. The region presents a continental-Mediterranean climate and the city is located in a natural basin
surrounded by Sierra Nevada mountains (up to 3500m a.s.l.). The city experiences a seasonal evolution of columnar aerosol
types, with larger AOD in summer and lower values in winter while the opposite occurs for AE (e.g., Alados-Arboledas
et al., 2003; Pérez-Ramírez et al., 2012a). The seasonal cycle in columnar aerosol properties is mostly associated with the
airmass pattern (Pérez-Ramírez et al., 2016) dominated by the more frequent and intense arrival of Saharan dust during summer
(e.g., Lyamani et al., 2006; Valenzuela et al., 2012; Antón et al., 2012; Román et al., 2013; Benavent-Oltra et al., 2017).
Anthropogenic aerosol sources in the region are mainly domestic heating and traffic (Lyamani et al., 2010; Titos et al., 2012).
Nevertheless, the region experiences in winter long periods of air masss stagnations that increase their pollution levels to values
compared with other European megacities (e.g., Casquero-Vera et al., 2019).

## 2.2   Instrumentation

The Sun/sky/Moon CE318-T photometer (*Cimel Electronique S.A.S.*) is used in this work to derive AOD at day and night-
time. This photometer is mounted on a two-axis robot and a tracking system allows measurements of direct solar and lunar
irradiance, and diffuse sky radiance at different geometries. The photometer head is mainly formed by a collimator, a filter
wheel (with narrow interference filters) and two detectors. The usual nominal wavelengths of the photometer filters are 340,
380, 440, 500, 675, 870, 935, 1020 and 1640 nm. The detectors are a Silicon sensor to measure the wavelengths of 1020 nm and
shorter, and an InGaAs sensor to measure the wavelengths equal or longer than 1020 nm; hence 1020 nm is measured by both
detectors. Sky at solar aureole and Moon measurements are recorded by the same detectors than Sun but with an amplification





gain around 128 and 4096 (formed by two amplifications of 128 and 32), respectively; the sky measurements out of the solar aureole are recorded with the same gain than Moon observations.

The CE318-T photometer (and older versions without the capability to observe the Moon) is the standard instrument in
AERONET. The photometers used in this paper belong to AERONET, being the #933 a reference photometer used at Izaña data, and the photometers #918 (from $16^{th}$ March 2016 to $25^{th}$ July 2016), #751 (from $25^{th}$ July 2016 to $26^{th}$ May 2017) and #788 (from $25^{th}$ May 2017 to $11^{st}$ October 2017) the ones operated at UGR station. These photometers were regularly calibrated following the AERONET protocols (Holben et al., 1998; Giles et al., 2019).

The star photometer EXCALIBUR (EXtinction CAmera and LumInance BackgroUnd Register; *Astronómica S.L.*) operated
at UGR station continuously from 2006 to 2011 and during special field campaigns since 2013. A detailed description of the star-photometer EXCALIBUR can be found in Pérez-Ramírez et al. (2008a, b). A brief overview is provided here. The star photometer EXCALIBUR largest innovation is the use of a CCD camera as detector attached to a commercial telescope of 30 cm diameter. A filter wheel permits the allocation of ten interference filters centered at 380, 436, 500, 532, 670, 880 and 1020 nm for aerosol studies, and an additional filter at 940 nm for precipitable water vapor measurement. In this work the 380, 436,
500, 670 and 880 nm channels are used. The one at 1020 could not be used due to technical problems. AOD in these spectral bands will be assumed equal to the AOD at 440, 500, 675 and 870 nm in order to compare with the CE318-T photometer. The AOD is computed from direct star irradiance using the one-star method, that is the same approach used for Sun photometry. The one-star method needs only a relative calibration of the instrument, but requires a first calibration for the entire set of stars used (Pérez-Ramírez et al., 2011). Nevertheless, a first calibration of the stars (isolated and stable stars) is enough as the
recalibration of the instruments consists only of computing wavelength dependent calibration factors that are the same for all the stars. Star photometer EXCALIBUR is able to provide measurements for all filters in approximately 1-2 minutes, but to minimize the effects of atmospheric turbulence data were averaged every 30 min (Pérez-Ramírez et al., 2011). A procedure based on moving averages an outlier removal is used for cloud-screening and data quality check (Pérez-Ramírez et al., 2012b). In addition, a visual inspection of data has been carried out to remove spurious data. Final uncertainties in AOD are 0.02 for
wavelengths below 800 nm and 0.01 for wavelengths above 800 nm (Pérez-Ramírez et al., 2011). Other authors reported a higher uncertainty in AOD from star photometry, about 0.02–0.03 (Baibakov et al., 2015; Barreto et al., 2019). The analysed period in this work is for coincident measurements of star and Moon photometer, and can be divided in two periods, in the framework of the SLOPE (Sierra Nevada Lidar AerOsol Profiling Experiment) I and II field campaigns (de Arruda Moreira et al., 2018; Bedoya-Velásquez et al., 2018; Casquero-Vera et al., 2020): from May $25^{th}$, 2016 to September $17^{th}$, 2016; and
from July $1^{st}$, 2017 to October $17^{th}$, 2017. Just before the second measurement period, EXCALIBUR was measuring at Izaña in the first multi-instrument nocturnal intercomparison campaign (Barreto et al., 2019).

## 2.3 Data management

The University of Valladolid (UVa; Spain) is in charge of one AERONET calibration center since 2006 and, in this framework, the UVa staff developed the CÆLIS software tool (Fuertes et al., 2017) with the aim of managing the data generated by
AERONET photometers and for calibration and quality control purposes. This tool contains relevant information about the





different photometers, like the spectral response of the filters or the signal temperature correction coefficients, and also includes climatology tables of different atmospheric variables (like pressure or the abundance of several absorption gases) useful to perform the atmospheric correction in the AOD calculation. An AOD calculation algorithm has recently been implemented in CÆLIS (González et al., 2020). Therefore, the day and night-time AOD data from the CE318-T measurements used in this work have been obtained from CÆLIS.

## 3 AOD from Moon observations

The main advantage of Sun photometry is that solar irradiance is directly emitted by the Sun and then, the solar irradiance reaching the top of atmosphere (extraterrestrial irradiance) does not significantly change, at least along one day, being the Earth-Sun distance the main factor modulating this irradiance and causing variations about $\pm 3\%$ along one full year. This fact makes that the knowledge of the absolute extraterrestrial irradiance is not needed in the AOD calculation, because an equivalent extraterrestrial irradiance, taking into account the Earth-Sun distance, can be obtained for a given instrument using Langley-plot or side-to-side calibration (Shaw, 1976, 1983; Holben et al., 1998; Toledano et al., 2018; Giles et al., 2019; González et al., 2020). However, the Moon is not a self-illuminating body but a diffuse solar reflector with exceptional stability (Kieffer and Stone, 2005); hence, lunar irradiance at the top of the Earth's atmosphere significantly changes, mainly with the Moon Phase Angle (MPA), even along one single night. This fact points out the need of knowledge of the extraterrestrial lunar irradiance for Moon photometry purposes. In this framework, AOD from lunar irradiance observation can be calculated following the Beer-Bouguer-Lambert law as follows (Barreto et al., 2013):

$$\tau_a(\lambda) = \frac{ln\left[\kappa^M(\lambda)\right] - ln\left[V^M(\lambda)/E_0^M(\lambda)\right] - m_g \cdot \tau_g(\lambda) - m_R \cdot \tau_R(\lambda)}{m_a} \tag{1}$$

where $\tau_a$ and $\kappa^M$ are the AOD and the Moon calibration coefficient, respectively, for a nominal $\lambda$-wavelength; $E_0^M$ and $V^M$ are the extraterrestrial lunar[1] irradiance and the photometer lunar signal at the same nominal $\lambda$-wavelength, respectively; while $m_a$, $m_R$ and $m_g$ are the optical airmass for aerosols, Rayleigh scattering and gaseous absorption, respectively, using the Moon Zenith Angle (MZA) instead of Solar Zenith Angle (SZA). Finally, $\tau_R$ and $\tau_g$ represent the optical depth of Rayleigh scattering and gaseous absorption, respectively. More details about these calculation in CÆLIS can be found in González et al. (2020).

### 3.1 Extraterrestrial Lunar Irradiance

As already mentioned, the knowledge of the extraterrestrial lunar irradiance is necessary in Moon photometry. To this end, the RIMO model has been implemented in CÆLIS. RIMO (http://testbed.aemet.es/rimoapp), which is described in detail in Barreto et al. (2019), is an implemetation of the ROLO model (Kieffer and Stone, 2005), which is mainly based on empirical relationships between the lunar irradiance measured at 32 channels by two CCD devices, both mounted in a telescope, and

---

[1]hereafter the superscripts $M$ and $S$ will make reference to Moon or Sun respectively.





the different geometrical factors of the Moon-observer positions. RIMO firstly calculates the reflectance of the Moon's disk
following the next equation (Eq. (12) in Barreto et al., 2019):

$$ln\left[A(k)\right]=\sum_{i=0}^{3}a_i(k)g_r^i+\sum_{j=1}^{3}b_i(k)\Phi^{2j-1}+c_1\phi+c_2\theta+c_3\Phi\phi+c_4\Phi\theta+d_1(k)exp\left(-\frac{g_d}{p_1}\right)+d_2(k)exp\left(-\frac{g_d}{p_2}\right)+d_3(k)cos\left(\frac{g_d-p_3}{p_4}\right)$$

(2)

where $A$ is the Moon's reflectance at one of the 32 $k$-wavelengths of the ROLO model; the $a$, $b$, $c$, $d$ and $p$ values are the
coefficients shown in Kieffer and Stone (2005); $g_r$ and $g_d$ are the absolute value of MPA in radians and in degrees, respectively;
$\Phi$ is the selenographic longitude of Sun (in radians); $\theta$ and $\phi$ are the selenographic latitude and longitude of the observer,
respectively, both in degrees (Barreto et al., 2019).

The Moon's reflectance $A$ is calculated by RIMO using equation (2) at the 32 ROLO wavelengths and, then, each one is
multiplied by a correction factor which was previously calculated by the comparison between a composite spectrum (95%
soil) of Moon's reflectance based on *Apollo 16* samples (soil and breccia) and the reflectance obtained with the equation (2),
assuming zero libration and $g = \Phi = 7º$ (see Barreto et al. 2019 for more details). The Moon's reflectance at any different
wavelength is obtained by linear interpolation of the $A$ calculated values. Finally, in order to obtain the lunar irradiance from
the Moon's reflectance, some geometric factors such as the distances between the Moon, the Sun and the observer must be
taken into account, as follows:

$$E_0^M(\lambda)=\frac{A(\lambda)\cdot\Omega_M\cdot E_0^S(\lambda)}{\pi}\left(\frac{1AU}{D_{S-M}}\right)^2\left(\frac{384400km}{D_{O-M}}\right)^2$$

(3)

where $E_0^M$ and $A$ are the extraterrestrial lunar irradiance and the Moon's reflectance, respectively, both at the $\lambda$-wavelength;
$E_0^S$ is the extraterrestrial solar irradiance at the $\lambda$-wavelength being obtained from Wehrli (1985) smoothed by a Gaussian
filter of 2 nm width; $\Omega_M$ is the solid angle of the Moon (6.4177E-5 sr); and $D_{S-M}$ and $D_{O-M}$ are the distances between the
Sun and the Moon (in AU) and between the observer and the Moon (in km), respectively. These distances, the MZA and all
the geometrical angles involved in equation (2) are obtained from the SPICE Toolkit (http://naif.jpl.nasa.gov/naif/toolkit.html)
(Acton Jr, 1996; Acton et al., 2018) developed by the NASA's Navigation and Ancillary Information Facility (NAIF). SPICE
is run using the planetary and lunar ephemeris *DE421* (Folkner et al., 2008) in addition to planetary constants kernel for the
Moon ($moon\_pa\_de421\_1900-2050.bpc$), and lunar frames kernel ($moon\_080317.tp$) (Seidelmann et al., 2007; Speyerer
et al., 2016); the SPICE kernels $pck00010.tpc$ and $naif0011.tls$ are also used for other planetary and time parameters. The
NAIF pinpoint tool is used to calculate the position of the observer in each station regarding the Mean Earth body-fixed
reference system ($MOON\_ME$).

## 3.2 Gain Calibration Method

Once the extraterrestrial lunar irradiance is obtained from geographical and time inputs, the AOD can be calculated at night-
time using equation (1) if the calibration coefficient $\kappa$ is known. Different methods are proposed in the literature for calibration





purposes (calculation of $\kappa$) since the accuracy of the Langley-plot method could be affected by the fast variations of the Moon illumination. One way is the so-called Lunar Langley calibration method (Barreto et al., 2013, 2016), which is similar to a

classic Langley-plot calibration but where the photometer signal is divided by the extraterrestrial lunar irradiance, as follows:

$$ln\left[\frac{V^M(\lambda)}{E_0^M(\lambda)}\right] = ln\left[\kappa^M(\lambda)\right] - m_a \cdot \left(\tau_a(\lambda) + m_R\frac{\tau_R(\lambda)}{m_a} + m_g\frac{\tau_g(\lambda)}{m_a}\right) \tag{4}$$

Under stable atmospheric conditions, $\kappa^M$ can be obtained from the y-intercept of a least square fit between $ln(V^M/E_0^M)$ and the aerosol optical airmass. However, the possible errors and uncertainties in $E_0^M$ are propagated to the value of $\kappa$ obtained by this method, although these uncertainties are partially masked in the AOD retrieval (equation (1)) because the $E_0^M$ values

are also used in the calculation. Recently, Barreto et al. (2017) found a dependence on MPA and MZA of the AOD calculated by this Lunar Langley method.

Another way to calculate $\kappa^M$ without the use of $E_0^M$ is by the so called Gain calibration method (Barreto et al., 2016). This method, based on a vicarious calibration, consists of transferring the calibration of the solar channels to the respective Moon ones. Both CE318-T detectors are the same for solar and lunar irradiance measurements. In order to reach a higher signal

range, the Moon signal is amplified, being multiplied by a gain factor, $G$. In fact, this factor is formed by 2 amplification steps, being the first one the Sun to solar aureole gain ($\approx$128) and the second one the solar aureole to Moon gain ($\approx$32). The nominal value of $G$ is therefore equal to 4096 ($2^{12}$). The values of $G$ were measured with an integrating sphere in the laboratory by Barreto et al. (2016) and Li et al. (2016). These authors found experimental values for $G$ differing less than 0.3% from the nominal value of 4096; hence, $G$ is assumed in CÆLIS as wavelength independent and with a constant value of 4096. Taking

into account that the only difference between Sun and Moon measurements is this Gain factor, the Sun calibration can be transferred to Moon as follows:

$$\kappa^M(\lambda) = \frac{V_0^S(\lambda)}{E_0^S(\lambda)} \cdot G \tag{5}$$

where $V_0^S$ is the Sun calibration coefficient and $E_0^S$ the extraterrestrial solar irradiance (Wehrli, 1985), both at the $\lambda$-wavelength. The Gain calibration is simpler than Lunar Langley method because it is not dependent on the RIMO (or other

lunar irradiance model) and it only requires the daytime calibration, which provides more operational character to this method.

### 3.3   RIMO Correction Factor

In order to evaluate the AOD obtained by the Gain calibration, the method of Barreto et al. (2017) has been followed, who assumed as a reference AOD, $AOD_{ref}$, the linear temporal interpolated values using the last daytime AOD value of the previous afternoon and the first AOD of the following morning, which make sense if stable and pristine conditions were

found during the night. Hence, the AOD obtained by the Gain calibration, equation (5) and (3) in equation (1), has been calculated for several nights that satisfied pristine and stable conditions to be compared against $AOD_{ref}$. Data from the #933 CE318-T photometer located at IZO have been selected for this purpose, since this high-elevation remote site usually presents





unique atmospheric conditions with very low and stable AOD values. The morning and afternoon solar Langley-plots from this photometer have been calculated, and stable conditions have been assumed when these Langley-plots present more than

25 data, the AOD at 500 nm below 0.025 and the standard deviation below 0.006 (see Toledano et al., 2018). The nights for which both the previous afternoon and the next morning solar Langley-plots fulfill the mentioned criteria, have been selected as the 'stable and pristine' nights. The AOD has been calculated for these selected nights but discarding optical airmasses larger than 6 and data under MPA absolute values above 90°. Moreover, some cloud contaminated nights have been discarded manually by visual inspection in order to warranty the AOD quality. As result, around 13500 AOD data points per wavelength,

corresponding to 98 pristine and stable nights from June 2014 to March 2018 at IZO, have been selected.

The differences between the AOD obtained by the Gain calibration and the reference values ($\Delta AOD_{G-r}$) have been calculated following the next equation:

$$\Delta \tau_{G-r}(\lambda) = \tau_{Gain}(\lambda) - \tau_{ref}(\lambda) \tag{6}$$

where $\Delta \tau_{G-r}$, $\tau_{Gain}$ and $\tau_{ref}$ are $\Delta AOD_{G-r}$, the AOD from the Gain calibration and the interpolated AOD used as

reference ($AOD_{ref}$), respectively, for the $\lambda$-wavelength. Figure 1a shows the obtained $\Delta AOD_{G-r}$ values as a function of the MPA at IZO for the 98 chosen stable nights and for all photometer channels. These differences point out negative values in the calculated AOD with Gain method and RIMO model, and the existence of a fictitious nocturnal cycle, symmetrical with the optical airmass, which could be associated in Sun photometry to a deficient calibration (Cachorro et al., 2004, 2008; Guirado et al., 2014). However, in Moon photometry this cycle, as equation (1) evidences, could be also caused by inaccuracies in

the used $E_0^M$ values. Barreto et al. (2017) found a similar behaviour in these differences but being close to zero for MPA≈0 and increasing with the absolute phase, which could be explained because they used Lunar Langley calibration near to the full Moon and it masked the possible bias on RIMO at least close to MPA≈0. Assuming the Gain calibration and $AOD_{ref}$ are right, and all the differences between AOD and the reference are caused by RIMO inaccuracies, the $\Delta AOD_{G-r}$ can be expressed as[2]:

$$\Delta \tau_{G-r}(\lambda) = \frac{1}{m} \cdot ln \left[ \frac{E_{0-ref}^M(\lambda)}{E_{0-RIMO}^M(\lambda)} \right] \tag{7}$$

where $E_{0-RIMO}^M$ is the extraterrestrial lunar irradiance from RIMO (the one used in CÆlis) and $E_{0-ref}^M$ is the extraterrestrial lunar irradiance that provides the $AOD_{ref}$ if the Gain calibration is applied, both for the $\lambda$-wavelength. A correction factor that transforms RIMO irradiance into the reference irradiance, named *RIMO Correction Factor* (RCF), is defined as the ratio between the extraterrestrial lunar irradiance assumed as reference and the obtained by RIMO. RCF can be derived for each

$\lambda$-wavelength from equation (7) as:

$$RCF(\lambda) = \frac{E_{0-ref}^M(\lambda)}{E_{0-RIMO}^M(\lambda)} = exp\left[m \cdot \Delta \tau_{G-r}(\lambda)\right] \tag{8}$$

---

[2]In order to simplify hereafter $\tau_a$ and $m_a$ are expressed as $\tau$ and $m$, respectively, without $a$ subscript.





The RCF values have been calculated by equation (8) using the data of Figure 1a, and they are shown in Figure 1b. The UV channels present high dispersion, while the longer wavelengths point out a decay in RCF close to the full Moon. The other channels show less dependence on MPA and, excluding the UV channels, the extraterrestrial lunar irradiance from RIMO
underestimates the assumed as reference between 1% and 14% (between 3% and 12% for MPA absolute values between 5º and 70º). This last result is in agreement with the differences reported by Lacherade et al. (2014), who found that ROLO underestimates around 6-12% (in the same MPA range than in this paper) for wavelengths between 505 and 844 nm, using as reference an imagery absolute calibrated system on board two PLEIADES satellites. Geogdzhayev and Marshak (2018) observed that ROLO underestimates, within 10%, the irradiance at six wavelengths between 443 and 780 nm using EPIC (Earth
Polychromatic Imaging Camera) images, calibrated using MODIS (Moderate Resolution Imaging Spectroradiometer) data. Viticchie et al. (2013) found also a positive bias around 15% between Moon observations from SEVIRI (Spinning Enhanced Visible and Infrared Imager) on-board MSG2 satellite (Meteosat Second Generation) and the ROLO model at 1600 nm, and a behaviour close to the full Moon similar to the observed in Figure 1b for the longer wavelengths. These independent results point out that ROLO, and hence its implementation RIMO, underestimates the extraterrestrial lunar irradiance, which is in
concordance with the obtained results and reinforces the hypothesis that the Gain calibration method is appropriate.

Viticchie et al. (2013) and Lacherade et al. (2014) observed a dependence of the differences between ROLO and satellite observations on MPA, and these dependencies on MPA are also observed in RIMO in Figure 1b. Uchiyama et al. (2019) used the Lunar Langley technique to observe an underestimation of the ROLO reflectance (given by equation (2)) with a MPA dependence fitted to a quadratic equation of the absolute value of the Moon phase angle ($C = A_c g^2 + B_c$; $g$= Moon phase
angle) compared with the reflectance obtained with photometer measurements; however, these authors did not consider the use of the solar spectrum of Wehrli (1985), which was used by Kieffer and Stone (2005) to derive the Moon reflectance of ROLO, neither asymmetries on the phase angle dependence of ROLO reflectance correction. Considering the results reported in the literature, RCF values of Figure 1b have been fitted by a least square method to a 2nd order polynomial as a function of MPA:

$$RCF(\lambda) = a(\lambda) + b(\lambda) \cdot g + c(\lambda) \cdot g^2 \qquad (9)$$

where $g$ is the MPA, and $a$, $b$ and $c$ the fitting coefficients at $\lambda$-wavelength. The obtained coefficients are shown in Table 1 for the different wavelengths; the RCF values produced by these coefficients are also shown in Figure 1c, indicating RCF values between 1.03-1.14 for all MPA range except for the UV channel, for which the fit indicates much larger dependence on MPA. The coefficients for 340 nm channel have been calculated only using data with MPA absolute values lower or equal to 55º since the AOD at 340 nm is too noisy due to the low lunar signal, especially far from the full Moon. The discrepancies in the RCF
value for 1020 nm between Silicon and InGaAs (1020i) channels (see Figure 1b) are also marked in the fitting coefficients which point out a RCF overestimation of InGaAs over Silicon around 0.03. The median (*MD*) and standard deviation (*SD*) of the RCF fitting residuals ($RCF_{resid}$) are also in Table 1, showing the worst fit for UV channels followed by 440 nm and the InGaAs channels; the InGaAs channels present a median and standard deviation in the RCF residuals around -0.001 and 0.013,





respectively, which could explain part of the mentioned discrepancies between the RCF values at 1020 nm in both Silicon and
InGaAs channels. On the other hand, the lowest deviation (around 0.01) is reached for the 675, 870 and 935 nm.

Unifying equation (1), (3), (5) and (9) with the coefficients of Table 1, the AOD can be calculated using the Gain calibration
method as:

$$\tau(\lambda) = \frac{1}{m} \cdot ln\left[\frac{V_0^S(\lambda)}{V^M(\lambda)} \cdot \frac{RCF(\lambda) \cdot A(\lambda)}{(D_{O-M} \cdot D_{S-M})^2} \cdot \frac{384400^2 \cdot \Omega_M \cdot G}{\pi}\right] - \frac{1}{m} \cdot [m_R \cdot \tau_R(\lambda) + m_g \cdot \tau_g(\lambda)] \qquad (10)$$

which is the final way used by CÆLIS to derive AOD at night-time, adding the RCF values and using the Gain method to
transfer Sun to Moon calibration. Finally, in order to see how the residuals in the RCF fitting are propagated to the AOD, the
median and standard deviation of the residuals between the AOD from equation (10) and the $AOD_{ref}$ in the 98 chosen stable
nights (used in the RCF fitting) are calculated and shown in Table 1. The highest AOD deviation appears for the UV channels,
especially for 340 nm (even taking into account that MPA absolute values above 55º has been discarded), being about 0.12. The
AOD deviations are below 0.01 for all channels above 400 nm, being the highest for 440 nm and the InGaAs channels (1020i
and 1640nm). As in RCF residuals, the lowest deviations are found in 675, 870 and 935 nm channels. These results point out
that the 340 nm channel (at least for MPA absolute values above 55º), and possibly 380 nm, should not be used due to the high
dispersion, which caused by the low signal to noise ratio of these channels. In addition, the AOD from InGaAs channels should
be carefully used since they present the highest deviation (apart from the UV channels). An example of the AOD at night-time
obtained by the proposed method using CÆLIS is shown in the Figure 8 of González et al. (2020), where the AOD continuity
from day to night-time can be appreciated for different sites and MPA values.

## 4   Moon vs star photometer

In order to evaluate the performance of the AOD calculated by the method developed in section 3, the AOD from a Moon
photometer has been compared with the AOD measured by a star photometer. To this end, the AOD from the different Moon
photometers at UGR station in 2016-2017 has been obtained from CÆLIS. These AOD data have been previously cloud-
screened using the criteria explained in González et al. (2020), which is similar to the one used at daytime by Giles et al.
(2019). The applied criteria are mainly based on: the recorded signal must be higher than a threshold value in some infrared
channel (to warranty the correct pointing at the Moon); the AOD variation in a triplet observation must be below a threshold;
the temporal variation of AOD at 500 nm must be smooth (below 0.01 per minute); among others. The AOD negative values, or
below a established threshold, have been not discarded in this work, since this kind of criteria are usually based on a threshold
marked by the AOD uncertainty, but in this case the uncertainty is still not well known. Finally the cloud-free AOD values
from Moon photometer have been averaged in 30 minutes intervals, for comparison purposes with the star photometer outputs,
that are 30 min averaged values (see Section 2.2).

Figure 2 shows AODs and AEs for day and night time for the Moon cycle (first to third quarter) in July 2016. Data presented
are from Sun photometry (daytime) and Moon and star photometry (night-time). Moon phase angle values are also provided.





Generally good day-to-night continuity is observed for different aerosol loads and MPA values. However AOD at 380 nm from Moon observations looks noisier, reaching high/low values at the beginning/end of the night (similar to a daytime calibration problem) for the lowest MPA values. The showed data period includes different aerosol episodes, such as Saharan desert dust outbreaks during 18-19$^{th}$ and 20-21$^{st}$ (both events studied by Benavent-Oltra et al., 2019 and Román et al., 2018); the presence of these coarse particles lead to a reduction in the AE values (calculated only if the 4 wavelengths between 440 and 870 are

available), which is also shown in Figure 2. The AE from Moon observations fits well between day and night-time even near the Moon quarters, but AE from star measurements presents more fluctuation especially from 22$^{nd}$ to 24$^{th}$ July 2016. These results point out the goodness of the AOD from Moon observations, except at 380 nm which is not used for AE calculations; however, the fluctuations in AE from star photometer could indicate some extra uncertainties or measurement issues in some channels.

Figure 3 shows 1:1 comparisons of Moon photometer AODs versus star photometer values for all data acquired during the intensive field campaigns. All channels show correlation between both AOD data sources, being the correlation coefficient, shown in Table 2, higher than 0.96 (0.97 if only 2016 is considered) except for 380 nm, which presents a lower value around 0.71. Table 2 also shows the slope and y-intercept of the linear fits shown in Figure 3, both ranging from 0.975 (440 nm) to 1.038 (870 nm) and from -0.012 (870 nm) to -0.004 (500 nm), respectively, for the wavelengths between 440 and 870 nm;

these results reveal that the obtained fitted lines are close to the 1:1 line. Table 2 also shows the mentioned statistical estimators calculated using only data of 2016 or 2017, separately. For the wavelengths between 440 and 870 nm, the correlation decreases to about 0.94 and the linear fits are farther than the 1:1 line for 2017. This worse relationship between both instruments in 2017 could be caused by some technical problems observed in the star photometer in 2017 after the participation of the instrument in the first multi-instrument nocturnal intercomparison campaign (Barreto et al., 2019) at Izaña, likely linked to the transport

of the instrument from Granada to Izaña and vice versa. In the case of the 380 nm, this channel presents higher agreement in 2017 than in 2016 due the large number of negative values of AOD from Moon observations registered in August and September 2016, especially during periods close to the Moon quarters. The AOD data these both months were derived from measurements recorded by the #751 photometer; AOD from this photometer also showed this behaviour for 380 nm for all the period of measurements at Granada in 2016 and 2017 (even out of SLOPE campaigns). These negative values cannot be

appreciated in Figure 3 since they are out of axis limits, and they are not cloud-screened since the used criteria does not reject negative values, but these negative values are the main cause of the shifted linear fit shown in Figure 3 for 380 nm, even when this graph shows a lot of AOD at 380 nnm values close to the line 1:1. In fact, if the agreement in the 380 nm channel is recalculated without the two mentioned months, the r coefficient, y-intercept and slope are 0.94, -0.03 and 0.97, respectively, using 309 data totally. The behaviour in the agreement of the other channels also shows a little improvement, but in this case

it is due to some negative AOD values from star photometer (although within the uncertainties) acquired in August 2016. The same statistical analysis has been done for the AE, showing worse agreement than the AOD. The AE agreement improves if the two troublesome months in 2016 are not included in the analyses. Actually, the improved analysis presents correlation coefficient of 0.79, slope of 0.85, and y-intercept of 0.20. However, removing the most problematic periods, the AE values do





not show as good agreement between both instruments as for the AOD, probably because individual deviations in AOD affects
AE computations, which is particularly critical for low AOD (Cachorro et al., 2008).

In order to quantify the discrepancies between the AOD retrieved by Moon and star photometers, Figure 4 shows frequency histograms of relative differences in AOD, assuming the star photometer as reference. Figure 4 reveals that in general the differences are centred around zero and normally distributed. The influence of the negative AOD at 380 from Moon observations can be observed in the negative tail showed by the differences in this channel distribution. The percentage of AOD
absolute difference values below 0.02 are 27%, 47%, 45%, 57% and 63% for 380, 440, 500, 675 and 870 nm, respectively; these percentages rise up to 46%, 65%, 63%, 69% and 75% for differences below 0.03. Table 2 shows the mean, median and standard deviation of the differences given in Figure 4. For the wavelengths between 440 and 870 nm, the mean and median of the differences are close to zero, being the absolute value below 0.01 except for 440 nm where the median in all measurement period is -0.012. These results point out that, for these wavelengths, there is no significant under or overestimation of AOD
from Moon to from star photometer, except a very small underestimation about 0.01 at 440 nm (within the uncertainty). The standard deviation, associated to the uncertainty, shows values about 0.04 for 440 to 675 nm and 0.03 for 870 nm; these values are reduced around 0.01 if they are calculated only with data from 2017, which can be due to the influence of the mentioned AOD values at August'16. If this month is removed from the dataset, then all the mentioned standard deviations go down to 0.03. The mean, median and standard deviation of the differences in the 380 nm channel are high, but significantly lower for
2017 likely due to the impact of negative AOD values from Moon observations obtained in the period August-September'16. The median and standard deviation are -0.03 and 0.06 when this period is removed. Regarding AE differences, the mean and median are below 0.10 for all data, indicating a lack of significant over or underestimation, but the standard deviation is around 0.4, revealing a high dispersion.

We have also investigated whether the performance of the AOD depends on the MPA, due to the influence of this parameter
on the incoming lunar irradiance and in the RCF values (see Section 3.3). Figure 5 shows the Moon-star AOD differences as a function of MPA for the different wavelengths. No dependence of the relative differences on MPA is observed, neither for the median values nor for the standard deviations. A high reduction in the differences can be observed for the 380 nm in the 70-80º MPA bin, which surely is the MPA bin with more negative AOD values from Moon observations at this wavelength as mentioned above. Finally, the AE differences do not show any clear pattern with MPA, but the high dispersion observed before
can be also appreciated.

## 5    Conclusions

Moon photometry needs accurate knowledge of the extraterrestrial lunar irradiance in order to calculate the aerosol optical depth (AOD). This paper uses the RIMO model (an implementation of the ROLO model) to calculate this irradiance, and a Sun/sky/Moon photometer (CE318-T) located at the high-altitude station of Izaña to take measurements of the lunar irradiance
at ground to derive the AOD. However, the AOD values obtained using these measurements and the RIMO model are not in agreement with the expected values even under pristine and clear conditions. The discrepancies between the obtained and the



expected AOD can be mainly caused by two issues: 1) bad calibration coefficients of the photometer or 2) lack of accuracy in the RIMO values. The calibration used in this work has been based on transferring the calibration of the solar channels (well established) to the Moon channel by a vicarious method, based on the fact that the photometer takes the Moon observations with the same sensor than Sun measurements but with a two-step electronic amplification of 4096 in the signal. In principle, nothing suggests that AOD errors could come from the calibration, while other works in the literature pointed out discrepancies in the ROLO model. This fact has motivated us to assume the lack of accuracy on RIMO as the responsible of the observed differences, and these differences have been used to determine the RIMO accuracy.

Detailed analyses of the differences between expected AOD and the AOD derived by RIMO have shown a bias revealing an underestimation of RIMO to the real extraterrestrial lunar irradiance about 1-14% for visible and IR channels, which in addition depends on the Moon phase angle (MPA); this result agrees with other works in the literature. The mentioned bias has been modeled as a function of MPA by a 2nd order polynomial (for each wavelength). These proposed polynomials represent the named RIMO correction factor (RCF), since if a RIMO irradiance output is multiplied by this factor, then the derived AOD from the corrected irradiance will be closer to the expected AOD. The obtained RCF values are at least useful for the retrieval of AOD from Moon observations. Differences around 0.03 in the RCF values has been found for the same wavelength (1020 nm) using two different detectors (Silicon and InGaAs); this result has apparently no physical sense, since the lunar irradiance cannot take two different values for one single wavelength. Consequently, this result must be caused by the uncertainty of the measurements and the method itself, indicating that the uncertainty of the estimated extraterrestrial lunar irradiance with RCF might be about 3%, at least for 1020 nm. The obtained results at 340 nm have been too noisy hence the use of this channel is not recommended. This new methodology based on the modeled RCF to correct RIMO for AOD calculation has been implemented in CÆLIS, achieving a night-time AOD calculation in near-real time for all photometers managed by this tool in an operational way. This is possible because the used calibration only needs from the routine Sun calibration (the so called Gain calibration method).

The RIMO-corrected AODs have been evaluated versus alternative and independent measurements from a star photometer. This instrument was deployed at Granada, a different location than the one used for the proposed RCF calculation. To our knowledge this is the first long-term AOD comparison between Moon and star observations. The obtained results for wavelengths between 440 and 870 nm have pointed out a good agreement between both databases, being the absolute mean difference below 0.01, except for 440 nm which is below 0.02. This indicates only a slight underestimation of AOD from Moon to star observations (used as reference) at 440 nm, but within the uncertainty of the star photometer (about 0.02-0.03). The standard deviation of the Moon-star AOD differences for the mentioned wavelengths is about 0.03-0.04, but if some problematic periods in the star photometer data are neglected, these values are reduced approximately to 0.01, which leads to an uncertainty in AOD from Moon observations between 0.019 (870 nm) and 0.028 (500 nm). However, these uncertainties could be lower because part of the observed differences could be caused by: detected technical problems in star photometer filter wheel; the differences in the effective wavelengths used in both instruments and in the way to correct atmospheric gaseous scattering and absorption at these wavelengths; the inhomogeneity of aerosol spatial distribution, since both instruments point to different targets, which also affects to the time interval used to the averages (clouds can block the Moon but not the pointed



star, and vice versa). The differences at 380nm are higher, showing in the best case an underestimation around 0.03 and an uncertainty about 0.06. These results suggest current limitations in using this channel, mainly caused by the low signal at this wavelength, which usually produces high dispersion and noisy AOD values close to the Moon quarters. Further improvements

and analyses need to be done in order to guarantee AOD quality in the UV region. The analyzed wavelengths have not shown any dependence on MPA in the Moon-star AOD comparison.This is an important result because it indicates that the proposed correction is able to remove any influence of the Moon cycle on the AOD.

The used night-time cloud-screening is in general the same that is used for daytime but without rejecting AOD values below a given threshold. In spite of providing apparently good results, the night-time cloud-screening is still in development and it

could change or add other specific criteria in the future due to the particularities of night-time measurements. The development of new cloud-screening criteria is out of the scope of this paper but, in the future, it could be based on the consideration of temporal smoothness in individual wavelengths or in the addition of a threshold value for the minimum acceptable AOD and for the minimum acceptable recorded signal per channel; this could help to warranty the AOD quality, especially in the noisier channels like 380 nm. Recently, AERONET is also providing AOD values from Sun/sky/Moon photometers with its

own method[3], but this product is still labeled as provisional at present, hence a direct comparison between the AOD from the proposed and the AERONET methods has not been considered.

To sum up, this work provides more evidences about the reported underestimation of RIMO/ROLO model to the real extraterrestrial lunar irradiance and points to the need for a correction of this model or the development of a new extraterrestrial lunar irradiance model, at least for accurate AOD calculation purposes. Meanwhile, at least until a more accurate lunar irra-

diance model is available, the proposed correction can help in providing AOD retrievals with the Moon. Moreover, additional studies using different Moon photometers or using alternative and independent night-time instrumentation, like lidar or star photometers, are highly recommended to characterize the AOD uncertainty of the proposed method.

*Data availability.* The used data are available from the authors upon request.

*Author contributions.* RR, RG and CT designed and developed the main concepts and ideas behind this work and wrote the paper with input

from all authors. RG, AB and RR implemented the RIMO model in CÆLIS. RR, JABO and DPR carried out the routine and calibration measurements of the star photometer. DPR computed and processed the AOD data from the star photometer. VEC, FJO, LAA and AMdF aided in interpreting the results and worked on the manuscript. All authors were involved in helpful discussions and contributed to the manuscript.

*Competing interests.* The authors declare that they have no conflict of interest.

---

[3]https://aeronet.gsfc.nasa.gov/new_web/Documents/Lunar_Algorithm_Draft_2019.pdf



*Acknowledgements.* The authors are grateful to the Spanish Ministry of Science, Innovation and Universities for the support through the ePOLAAR project (RTI2018-097864-B-I00). This work was also supported by the Spanish Ministry of Economy and Competitiveness through projects CGL2016-81092-R, and CGL2017-90884-REDT; by the Andalusia Regional Government through project P18-RT-3820; and by the European Union's Horizon 2020 research and innovation program through ACTRIS-IMP (grant agreement No 871115). We thank Emilio Cuevas and their staff for establishing and maintaining the Izaña station used in this investigation.



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



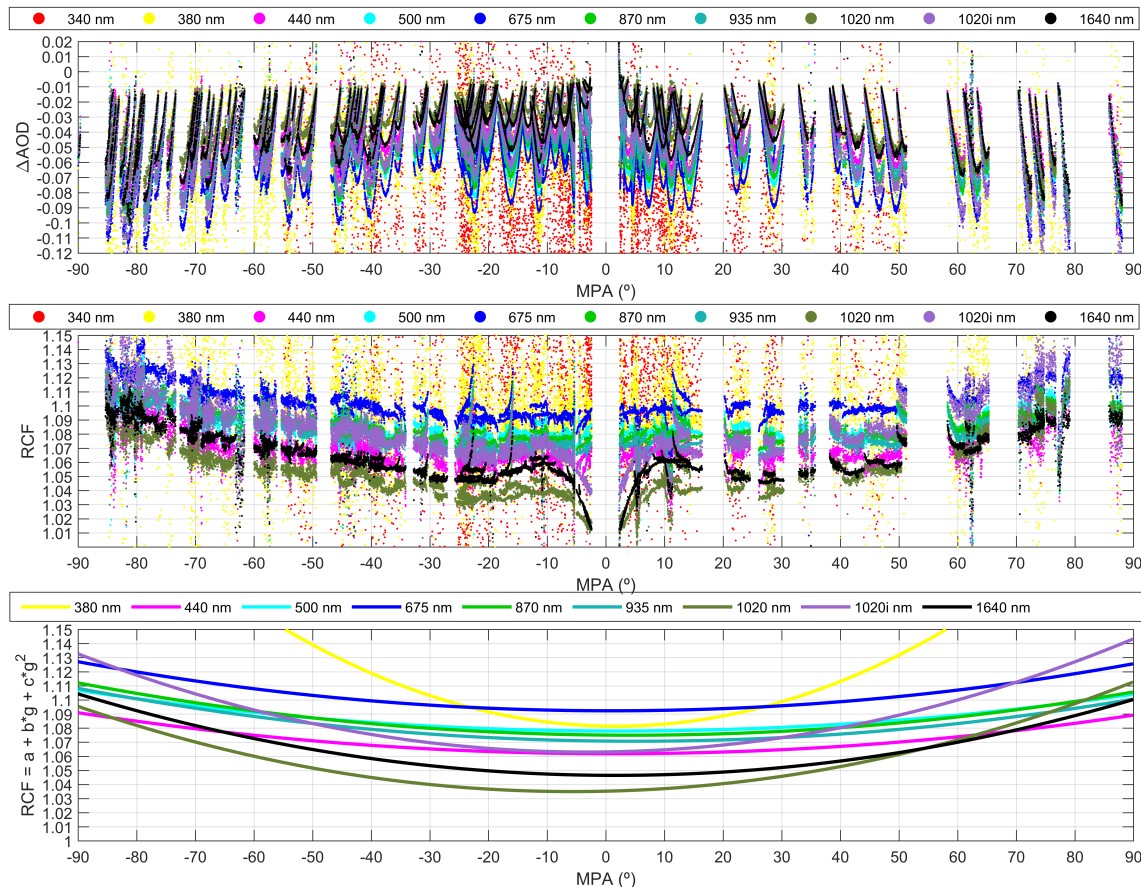

**Figure 1.** a) Differences between AOD from Gain calibration and the reference values at night as function of Moon Phase Angle (MPA) for different wavelengths; b) RIMO Correction Factor (RCF) against MPA for different wavelengths; and c) Fitted RCF against MPA for different wavelengths (340 nm values are not shown because they are out of the axis limits).

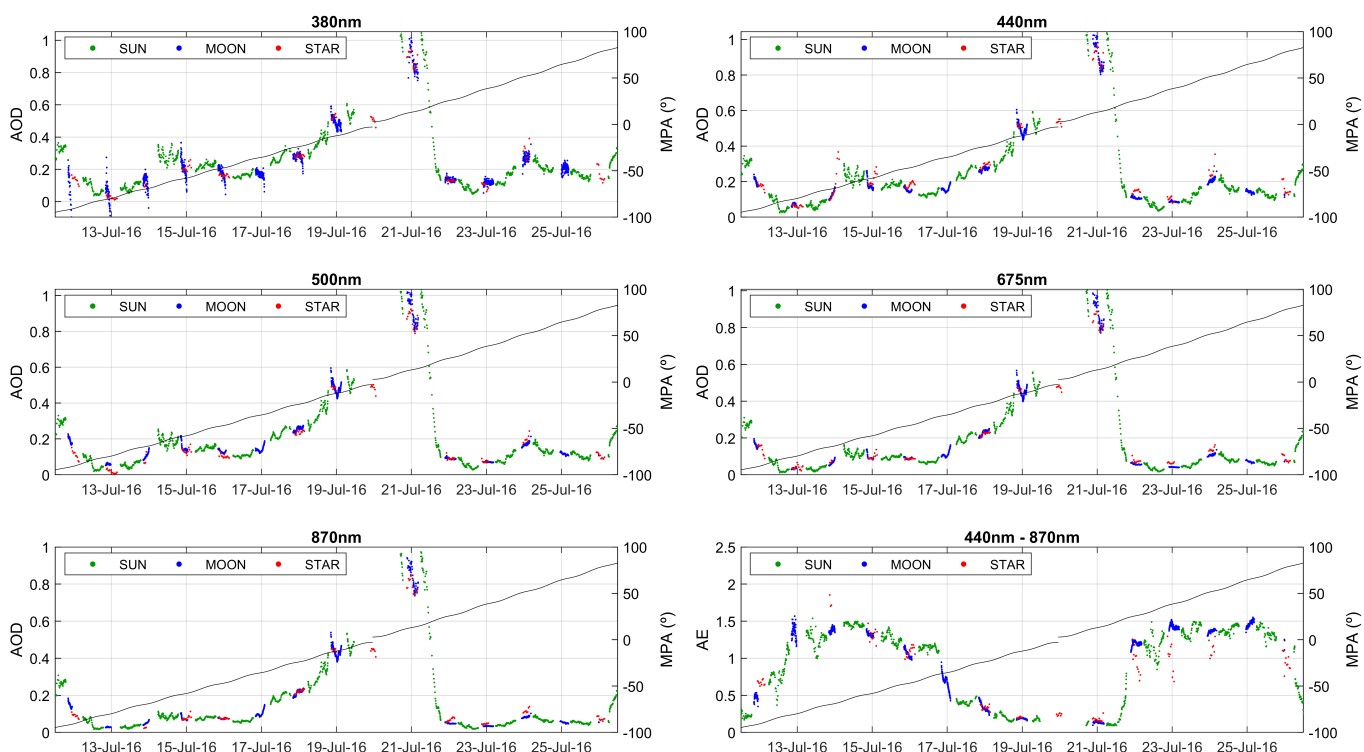

**Figure 2.** Aerosol optical depth (AOD) values from Sun, Moon and star photometer at Granada (Spain) from the first to third Moon quarter in July 2016. Bottom panel at right shows the Ångström Exponent (AE) calculated with the wavelengths of 440, 500 and 675 and 870 nm (436, 500, 670 and 880 nm for star photometer). Moon phase angle (MPA) is represented with a black line in each panel.

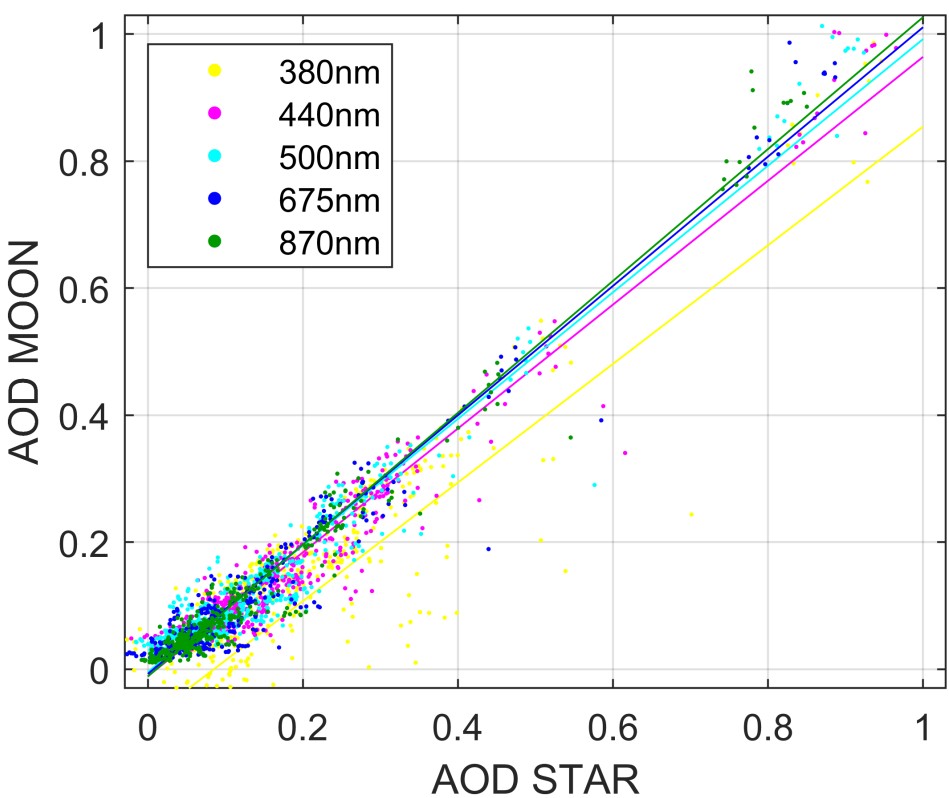

**Figure 3.** Aerosol optical depth (AOD) from Moon photometer versus the AOD from star photometer for different wavelengths. Linear fit line is also represented for each wavelength.



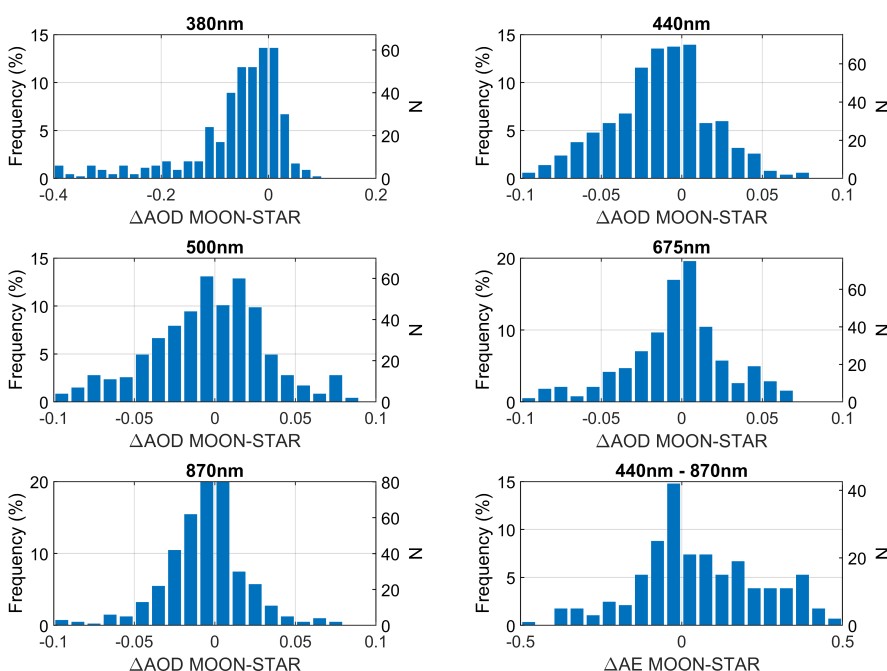

**Figure 4.** Frequency of the aerosol optical depth (AOD) differences between the Moon and star photometers for different wavelengths. Bottom panel at right shows the frequency of these differences but for Ångström Exponent (AE) in the 440-870 nm range.



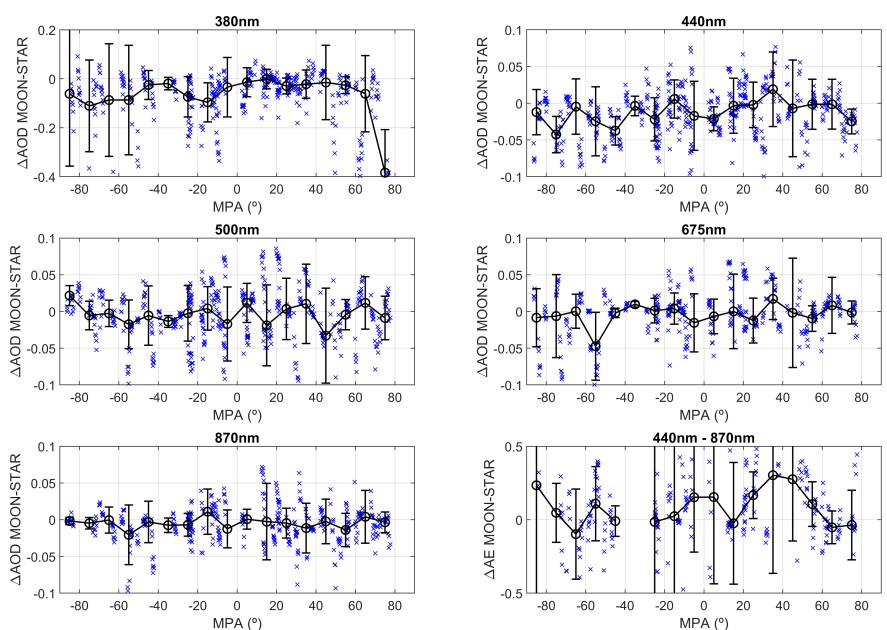

**Figure 5.** Aerosol optical depth (AOD) differences between the Moon and star photometers as a function of Moon phase angle (MPA) for different wavelengths. Bottom panel at right shows these differences but for Ångström Exponent (AE) in the 440-870 nm range. Black circles represents the median of all differences in a ±5º MPA interval, while the error bars are ± the standard deviation of the data in the same interval.





**Table 1.** Fitting coefficients of the RIMO Correction Factor (equation (8)), the number of used data ($N$), and median ($Md$) and standard deviation ($SD$) of the residuals in RIMO Correction Factor (RCF) and aerosol optical depth (AOD) for different photometer wavelengths. The fitting values at 340 nm has been obtained without MPA absolute values above 55º.

| $\lambda(nm)$ | N | a | b $(rad^{-1})$ | c $(rad^{-2})$ | $Md(RCF_{resid})$ | $SD(RCF_{resid})$ | $Md(AOD_{resid})$ | $SD(AOD_{resid})$ |
|---|---|---|---|---|---|---|---|---|
| 340 | 8895 | 1.186 | -2.35e-02 | 1.92e-01 | 6.15e-02 | 4.89e-01 | 3.42e-02 | 1.21e-01 |
| 380 | 13447 | 1.082 | -4.17e-03 | 7.10e-02 | 4.41e-03 | 1.70e-01 | 2.46e-03 | 5.37e-02 |
| 440 | 13496 | 1.062 | -5.35e-04 | 1.14e-02 | -4.71e-04 | 1.59e-02 | -2.41e-04 | 8.23e-03 |
| 500 | 13496 | 1.078 | -8.93e-04 | 1.11e-02 | -2.71e-04 | 1.28e-02 | -1.38e-04 | 6.88e-03 |
| 675 | 13496 | 1.092 | -4.50e-04 | 1.38e-02 | -1.77e-04 | 1.13e-02 | -8.77e-05 | 6.06e-03 |
| 870 | 13496 | 1.075 | -2.05e-03 | 1.37e-02 | -3.00e-04 | 1.12e-02 | -1.53e-04 | 6.17e-03 |
| 935 | 13494 | 1.071 | -2.41e-03 | 1.36e-02 | -2.29e-04 | 1.12e-02 | -1.21e-04 | 6.24e-03 |
| 1020 | 13495 | 1.035 | 5.55e-03 | 2.79e-02 | -2.36e-04 | 1.32e-02 | -1.18e-04 | 7.78e-03 |
| 1020i | 13495 | 1.063 | 3.40e-03 | 3.04e-02 | -7.35e-04 | 1.35e-02 | -3.63e-04 | 8.09e-03 |
| 1640 | 13495 | 1.047 | -1.25e-03 | 2.26e-02 | -4.38e-04 | 1.27e-02 | -2.25e-04 | 8.09e-03 |





**Table 2.** Statistical estimators of the differences between the aerosol optical depth (AOD) from Moon and star photometers for different wavelengths and periods. *N* is the number of used data; *M*, *Md* and *SD* represents the mean, median and standard deviation of these differences, respectively; $y_0$, *slp* and *r* are the y-intercept, slope and correlation coefficient from the linear fit between the AOD from Moon and star photometers. These estimators are also presented for the Ångström Exponent (AE) in the 440-870 nm range.

| $\lambda(nm)$ | $Period$ | $N$ | $M$ | $Md$ | $SD$ | $y_0$ | $slp$ | $r$ |
|---|---|---|---|---|---|---|---|---|
| | 2016 | 265 | -0.122 | -0.048 | 0.181 | -0.114 | 0.959 | 0.714 |
| 380 | 2017 | 183 | -0.051 | -0.040 | 0.062 | -0.001 | 0.762 | 0.787 |
| | All | 448 | -0.093 | -0.044 | 0.149 | -0.080 | 0.934 | 0.710 |
| | 2016 | 336 | -0.013 | -0.009 | 0.043 | -0.010 | 0.979 | 0.974 |
| 440 | 2017 | 166 | -0.019 | -0.014 | 0.027 | -0.008 | 0.938 | 0.946 |
| | All | 502 | -0.015 | -0.012 | 0.038 | -0.011 | 0.975 | 0.971 |
| | 2016 | 304 | 0.006 | 0.008 | 0.040 | 0.005 | 1.007 | 0.978 |
| 500 | 2017 | 162 | -0.025 | -0.024 | 0.031 | -0.013 | 0.926 | 0.918 |
| | All | 466 | -0.005 | -0.003 | 0.040 | -0.004 | 0.997 | 0.969 |
| | 2016 | 315 | -0.001 | 0.002 | 0.039 | -0.004 | 1.020 | 0.979 |
| 675 | 2017 | 68 | -0.021 | -0.020 | 0.032 | -0.031 | 1.061 | 0.934 |
| | All | 383 | -0.005 | -0.001 | 0.038 | -0.007 | 1.018 | 0.976 |
| | 2016 | 264 | -0.006 | -0.003 | 0.034 | -0.011 | 1.038 | 0.986 |
| 870 | 2017 | 137 | -0.009 | -0.008 | 0.024 | -0.012 | 1.027 | 0.939 |
| | All | 401 | -0.007 | -0.005 | 0.031 | -0.012 | 1.038 | 0.983 |
| | 2016 | 221 | 0.06 | 0.01 | 0.45 | 0.52 | 0.51 | 0.683 |
| AE(440-870) | 2017 | 63 | 0.15 | 0.11 | 0.28 | 0.11 | 1.06 | 0.656 |
| | All | 284 | 0.08 | 0.04 | 0.42 | 0.46 | 0.56 | 0.693 |