# Peer review of "Correction of a lunar irradiance model for aerosol optical depth retrieval and comparison with star photometer"

_Atmospheric Measurement Techniques, 2020_

## Referee Comment (RC1) · Anonymous Referee #1 · 1 Sep 2020

In this paper, a method for estimating RIMO correction factor (RCF) was developed for correcting the low bias in lunar irradiance as computed from the RIMO model. The RCF was developed by comparing reference aerosol optical depth (AOD) values estimated using daytime observations over pristine conditions with AODs estimated from Gain calibration. The retrieved nighttime AODs from moon photometer, with the use of RIMO RCF, are inter-compered with AODs from star photometer measurements. This paper presents a study that shall be interesting to users who use RIMO for nighttime aerosol and cloud property retrievals. Still, there are some minor issues that I would like the authors to make changes.

[Figure]

Line 39-41, "Finally, the direct effect of aerosols on solar radiation at night-time is avoided, but the aerosols presented at night-time can profoundly modify the longwave balance by means of the change in cloud properties and the impact on the longwave radiation absorbed by clouds, which is back-emitted to the Earth's surface". Add references to justify the comment.

Line 121, "Sky at solar aureole and Moon measurements are recorded by the same detectors than Sun but with an amplification" This sentence is confusing and may need to be rewritten. What is "same detectors than Sun"?

Line 123, "are recorded with the same gain than Moon observations". I believe "than" should be "as"?

Lines 135-136, "AOD in these spectral bands will be assumed equal to the AOD at 440, 500, 675 and 870 nm in order to compare with the CE318-T photometer" Why not interpolate star photometer data to the precise wavelengths as used by the moon photometer. If the authors do not want to match wavelengths from the two instruments, they need to document uncertainties introduced by the differences in wavelengths between the two instruments.

Lines 164-165, "This fact makes that the knowledge of the absolute extraterrestrial irradiance is not needed in the AOD calculation" This sentence is confusing. Please rewrite

Line 213, please explain the "Langley-plot method" using a few sentences. Not all readers know about the concept.

Lines 248-249, "Moreover, some cloud contaminated nights have been discarded manually by visual inspection in order to warranty the AOD quality." What are the criteria for the mentioned visual inspection? Home many data points are excluded by this step?

Lines 257-259, "These differences point out negative values in the calculated AOD with Gain method and RIMO model, and the existence of a fictitious nocturnal cycle,

symmetrical with the optical airmass, which could be associated in Sun photometry to a deficient calibration" This sentence doesn't make sense. "point out" should be "suggests that"??

Lines 262-263, "Assuming the Gain calibration and AODref are right," What do authors mean by "right"? I assume that the authors want to say that "Assuming the Gain calibration and AODref are accurate"??,

Line 298, "MPA absolute values lower or equal to 55° since" Any reason for picking 55 degree as the threshold?

Line 345, What is the study period for Figure 3?

Line 345, for a comparison purpose, can the authors also add a plot that is similar to Figure 3 but without using the RIMO RCF (e.g. using the original RIMO model)?

Lines 359-360, what are the causes of the negative values? Can figure 3 be modified to include negative AOD values?
* * *

---

## Referee Comment (RC2) · Anonymous Referee #2 · 19 Sep 2020

The paper points out the importance of the accurate knowledge of the Moon extraterrestrial spectrum over a full moon cycle for nighttime AOD retrievals in lunar photometry. A large dataset of Langley extrapolated values at Cimel's photometer wavelengths, covering the spectral region 380 nm -1640 nm, has been retrieved under stable and low AOD conditions, leading to an empirical spectral correction factor (RCF) of the RIMO model with respect to MPA. The number of data points and the ideal conditions is expected to lead to a low uncertainty correction factor. The validation of the RCF, by AOD comparison of Cimel photometer against a star photometer gives convincing results always within the uncertainties of the two independent retrievals. I find this work very interesting as it leads to a very useful and practical correction that allows nighttime

AOD retrievals based on the lunar photometry, in anticipation of a traceable update of ROLO and RIMO models.

Comments

1. The correction methodology described in the paper in based on the assumption of linear behavior of the instrument with respect to the measured irradiance. The authors need to address this in the paper, to avoid any confusion between instrumental and RIMO correction.

2. What is the spectral uncertainty of the correction? Figure 1 should include a panel demonstrating the uncertainty with respect to MPA as well as the relative RCF to a selected MPA.

3. Has the RCF been applied to other photometers/spectroradiometers?

4. How the degradation of the reference Cimel is accounted for? Are the daytime calibrations used between the night observation?

5. The stability of the atmospheric aerosol load has been well described, however what is maximum difference between the afternoon and next morning AOD to retrieve the correction factor? Is there any dependency of the RCF to the slope of the linear fit?

6. Apart from the comparison of the corrected AOD to the star photometer it would be very interesting to add in figures 2,3,4 the uncorrected AOD retrievals, so the reader can visualize the improvement.

7. A spectral RCF version of the Figure 1c for selected MPA would be helpful.

8. Why the could-flagging is wavelength dependent? Given the noise of 380 nm why the cloud flag from next measured wavelength is not used?

Technical comments/suggestions

Line 2: that is very relevant in polar areas Important, interesting, high value

Line 14: that provides the expected AOD values provides AOD closer to the expected values

Line 87: located below the Izaña's level. located below Izaña's level /altitude.

Line 121: same detectors as the Sun Line 125: the photometers used in this paper belong to AERONET, being the #933 a reference photometer used at Izaña data Used for Izaña data / operated at Izaña What is the measurement period?

Line 165: makes that the knowledge of the absolute extraterrestrial irradiance is not needed in the AOD calculation, because an equivalent Noncompulsory

Line 167: calibration transfer Line 170: this fact points out the need of knowledge of the extraterrestrial lunar irradiance for Moon photometry purposes this fact points out the need of knowledge of the extraterrestrial lunar irradiance, and especially the variation with respect to the MPA, for Langley based Moon photometry purposes

Line 360: appreciated in Figure 3 since they are out of axis limits, and they are not cloud-screened since the used criteria does not reject Seen

---

## Author Comment (AC1) · 8 Oct 2020

**Response to the Referee #1 comments for the manuscript "Correction of a lunar irradiance model for aerosol optical depth retrieval and comparison with star photometer" By Roberto Román et al. in AMTD**

First, we are grateful for the effort of Referee #1 and her/his review in detail. Reviewer comments are in black font (RC), and author comments (AC) in red font.

**Author's answer to Anonymous Referee #1**

RC: In this paper, a method for estimating RIMO correction factor (RCF) was developed for correcting the low bias in lunar irradiance as computed from the RIMO model. The RCF was developed by comparing reference aerosol optical depth (AOD) values estimated using daytime observations over pristine conditions with AODs estimated from Gain calibration. The retrieved nighttime AODs from moon photometer, with the use of RIMO RCF, are inter-compered with AODs from star photometer measurements. This paper presents a study that shall be interesting to users who use RIMO for nighttime aerosol and cloud property retrievals. Still, there are some minor issues that I would like the authors to make changes.

Line 39-41, "Finally, the direct effect of aerosols on solar radiation at night-time is avoided, but the aerosols presented at night-time can profoundly modify the longwave balance by means of the change in cloud properties and the impact on the longwave radiation absorbed by clouds, which is back-emitted to the Earth's surface". Add references to justify the comment.

AC: The sentence has been modified in order to be clearer and some references have been added. The new sentence is:

*"Moreover, the aerosols at night-time can profoundly modify the longwave balance by means of the change in cloud properties, such as cloud lifetime, and the impact on the longwave radiation absorbed by clouds which is back-emitted to the Earth's Surface (Ramanathan et al., 1989; Boucher et al., 2013)."*

References:

Ramanathan, V., Cess, R. D., Harrison, E. F., Minnis, P., Barkstrom, B. R., Ahmad, E., and Hartmann, D.: Cloud-Radiative Forcing andClimate: Results from the Earth Radiation Budget Experiment, Science, 243, 57–63, https://doi.org/10.1126/science.243.4887.57, 1989.

Boucher, O., Randall, D., Artaxo, P., Bretherton, C., Feingold, G., Forster, P., Kerminen, V.-M., Kondo, Y., Liao, H., Lohmann, U., et al.:Clouds and aerosols, in: Climate change 2013: the physical science basis. Contribution of Working Group I to the Fifth Assessment Reportof the Intergovernmental Panel on Climate Change, pp. 571–657, Cambridge University Press, 2013.

RC: Line 121, "Sky at solar aureole and Moon measurements are recorded by the same detectors than Sun but with an amplification" This sentence is confusing and may need to be rewritten. What is "same detectors than Sun"?

AC: The photometer takes measurements of solar irradiance (Sun measurements) at daytime. This Sun measurements are taken with two detectors (one Silicon detector one InGaAs detector for the short wave infrared wavelengths), but without any amplification. Both detectors are also used for the sky and Moon measurements, but due to the lower signal in these cases, electronic amplification is used to increase the signal to noise ratio. The gain factor is 128 for aureole (sky) measurements and 4096 for direct Moon observations. We have rewritten the sentence as follows:

*"Sky radiance at solar aureole and direct Moon irradiance are measured with the same detectors used to measure direct solar irradiance, but with an electronic amplification factor (gain) of 128 and 4096, respectively."*

RC: Line 123, "are recorded with the same gain than Moon observations". I believe "than" should be "as"?

AC: This sentence has been modified (see previous comment) and the word "than" has been removed.

RC: Lines 135-136, "AOD in these spectral bands will be assumed equal to the AOD at 440, 500, 675 and 870 nm in order to compare with the CE318-T photometer" Why not interpolate star photometer data to the precise wavelengths as used by the moon photometer. If the authors do not want to match wavelengths from the two instruments, they need to document uncertainties introduced by the differences in wavelengths between the two instruments.

AC: The wavelength differences between both instruments are small, being 4, 5 and 10 nm for 440 nm (436 nm for star ph.), 675 nm (670 nm for star ph.) and 870 nm (880 nm for star ph.) nm. To study the influence of this assumption, the AOD values from Moon photometer have been interpolated to the star photometer wavelengths following the Angström law (using the two neighbour wavelengths). The changes on the obtained results (Table 2) are not significant when this interpolation is applied.

Moreover, the mean values of the AOD differences with and without interpolation for the analysed data have been -0.0013, -0.0007 and 0.0006 for $AOD_{440}$-$AOD_{436}$, $AOD_{675}$-$AOD_{670}$ and $AOD_{870}$-$AOD_{880}$, respectively. These differences are one order of magnitude below the daytime AOD uncertainty.

Hence, we decided not to interpolate AOD to match the wavelengths, because it does not affect significantly the obtained results. The wavelengths are really close and we prefer to use the measured AOD by each instrument in order to avoid any kind of possible artefact caused by the wavelength interpolations.

In the text it is mentioned now:

*"AOD in these spectral bands will be compared to the AOD at 440, 500, 675 and 870 nm of the CE318-T photometer; the central wavelengths of these bands are close enough (below 10 nm difference) to allow a direct comparison of measured AOD and avoid interpolated data. If AOD of the CE318-T is interpolated to match the star photometer bands, the comparison does not significantly change (in general AOD differences below 0.001)."*

RC: Lines 164-165, "This fact makes that the knowledge of the absolute extraterrestrial irradiance is not needed in the AOD calculation" This sentence is confusing. Please rewrite

AC: It has been rewritten as:

"*A main advantage of Sun photometry is that the measured irradiance is directly emitted by the Sun and then, the solar irradiance reaching the top of atmosphere (extraterrestrial irradiance) does not significantly change, at least along one day. The Earth-Sun distance is the main factor modulating this irradiance, causing variations about ±3% along the year. Following the Beer-Bouguer-Lambert law, the extraterrestrial signal of the instrument (rather than irradiance in physical units) is needed for AOD calculation. This can be obtained by the Langley plot method (Shaw, 1976, 1983), in which direct Sun irradiance is observed at different solar elevations in order to extrapolate the top-of-the-atmosphere signal of the instrument. Side by side comparison with a reference instrument is the common practice in AERONET for calibration transfer in field instruments (Holben et al., 1998; Toledano et al., 2018; Giles et al., 2019; González et al., 2020).*"

RC: Line 213, please explain the "Langley-plot method" using a few sentences. Not all readers know about the concept.

AC: See comment above.

RC: Lines 248-249, "Moreover, some cloud contaminated nights have been discarded manually by visual inspection in order to warranty the AOD quality." What are the criteria for the mentioned visual inspection? Home many data points are excluded by this step?

AC: This step rejected 37 nights. This information is included in the revised manuscript:

"*Moreover, a total of 37 cloud contaminated nights have been manually discarded by visual inspection (nights without a smooth AOD time series) in order to warranty the AOD quality*"

The rejection of these nights is important, since the pristine conditions can be achieved at the previous afternoon and the next morning but clouds can appear during the night, producing non useful data. The best way to detect cloud contaminated data is by data visualization (AOD spikes, etc., see plots below). A standard cloud-screening may not properly work because uncorrected RIMO values produce unrealistic AOD: negative values, strong dependence on moon zenith angle, reverse wavelength dependence, etc. Such non-physical AOD would be rejected by any screening algorithm, even in clear nights.

When a night is selected as pristine (it previously satisfied the established Langley criteria in the previous afternoon and the next morning) and cloud-free, the behaviour of AOD is the shown in the example of Figure R1 for 500 nm. The AOD at night-time does not present good values (dependence on MZA and negative AOD) since the RCF is not applied, but its time series is smooth, which indicates no cloud presence. However, Figure R2 shows a case satisfying the pristine conditions before and after the night but with cloud

presence during the night, which is appreciated in the data jumps and the non smooth AOD time series. The case of Figure R2 is one of the 37 rejected nights.

[Figure]

Figure R1: Aerosol Optical Depth (AOD) at 500 nm at daytime and night-time without RCF correction using Gain calibration method at Izaña (Spain) from 19 to 20 August 2016.

[Figure]

Figure R2: Aerosol Optical Depth (AOD) at 500 nm at daytime and night-time without RCF correction using Gain calibration method at Izaña (Spain) from 16 to 17 August 2017.

RC: Lines 257-259, "These differences point out negative values in the calculated AOD with Gain method and RIMO model, and the existence of a fictitious nocturnal cycle, symmetrical with the optical airmass, which could be associated in Sun photometry to a deficient calibration" This sentence doesn't make sense. "point out" should be "suggests that"??

AC: The sentence has been modified as:

*"These differences show negative values, which is because the calculated AOD with Gain method and RIMO model is mostly below zero. A fictitious nocturnal cycle, symmetrical with the optical airmass, appears in these differences, and hence in the calculated AOD*

*with Gain method and RIMO; this kind of fictitious cycle are usually associated in Sun photometry to a deficient calibration (Cachorro et al., 2004, 2008; Guirado et al., 2014)*"

RC: Lines 262-263, "Assuming the Gain calibration and AODref are right," What do authors mean by "right"? I assume that the authors want to say that "Assuming the Gain calibration and AODref are accurate"??,

AC: The referee is right. We have replaced "*right*" by "*accurate*" in the new manuscript version.

RC: Line 298, "MPA absolute values lower or equal to 55 since" Any reason for picking 55 degree as the threshold?

AC: This threshold was based on the observation of RCF data. 340 nm is too noisy, but the dispersion is even greater for MPA absolute values above 55°, where the low lunar irradiance signal makes the signal to noise ratio too low. Anyway, we recommend that this channel is not used.

RC: Line 345, What is the study period for Figure 3?

AC: The period encompasses 2016 and 2017, it is the full period with star photometer data used in this work. This has been added in the Figure caption:

"Figure 3: *Aerosol optical depth (AOD) and Angström Exponent (AE) from Moon photometer versus the AOD and AE from star photometer for 2016-2017 period and for different wavelengths. Colour legend represents the relative density of data points. Black lines indicate linear fit to the data*"

RC: Line 345, for a comparison purpose, can the authors also add a plot that is similar to Figure 3 but without using the RIMO RCF (e.g. using the original RIMO model)?

AC: Figure 1 already shows that AOD presents wrong values when RCF is not applied. Hence, we do not consider that a similar analysis but with data that we know is wrong helps to improve the paper clarity. In addition, the automatic AOD cloud-screening will not work well with these data and we would need to transfer the cloud-screening results of the AOD calculated with RCF to the AOD without RCF correction. Nevertheless, we have done the comparison and it is shown in the Figure R3. As can be observed, the AOD from Moon underestimates about 0.05 the AOD from stars if the RCF correction is not applied. It is clearer in the Figure R4, where the differences are represented as a function of the MPA.

[Figure]

Figure R3: Aerosol optical depth (AOD) from Moon photometer with and without RCF correction versus the AOD from star photometer for 2016-2017 period and for different wavelengths. Linear fits are also represented for each wavelength.

[Figure]

Figure R4: Aerosol optical depth (AOD) differences between the Moon and star photometers as a function of Moon phase angle (MPA) for different wavelengths. Bottom-right panel shows these differences for Angström Exponent (AE) in the 440-870 nm range. Black circles represent the median of all differences in a ±5° MPA interval, while error bars indicate ± standard deviation of the data in the same interval.

RC: Lines 359-360, what are the causes of the negative values? Can figure 3 be modified to include negative AOD values?

AC: The manuscript explains that there are some problems with the 380 nm channel. Moon irradiance is low and then this channel is noisy with a low signal to noise ratio, especially for high MPA values. In fact, the paper recommends not to use of this channel for these reasons. The other wavelengths do not show these negative values; hence, the Figure 3 has been divided in 6 panels (one per wavelength and one more for AE) in the new manuscript (see Figure R5). Now the negative values in the 380 nm channel are shown while the other channels maintain the previous axis limits, so that all data are displayed. In addition, the scatter plots have been replaced by density scatter plots, adding the density of data points through a colour legend.

[Figure]

Figure R5: Aerosol optical depth (AOD) and Angström Exponent (AE) from Moon photometer versus the AOD and AE from star photometer for 2016-2017 period and for different wavelengths. Colour legend represents the relative density of data points. Black lines indicate linear fit to the data.

---

## Author Comment (AC2) · 8 Oct 2020

**Response to the Referee #2 comments for the manuscript "Correction of a lunar irradiance model for aerosol optical depth retrieval and comparison with star photometer" By Roberto Román et al. in AMTD**

First, we are grateful for the effort of Referee #1 and her/his review in detail. Reviewer comments are in black font (RC), and author comments (AC) in red font.

**Author's answer to Anonymous Referee #1**

RC: The paper points out the importance of the accurate knowledge of the Moon extraterrestrial spectrum over a full moon cycle for nighttime AOD retrievals in lunar photometry. A large dataset of Langley extrapolated values at Cimel's photometer wavelengths, covering the spectral region 380 nm -1640 nm, has been retrieved under stable and low AOD conditions, leading to an empirical spectral correction factor (RCF) of the RIMO model with respect to MPA. The number of data points and the ideal conditions is expected to lead to a low uncertainty correction factor. The validation of the RCF, by AOD comparison of Cimel photometer against a star photometer gives convincing results always within the uncertainties of the two independent retrievals. I find this work very interesting as it leads to a very useful and practical correction that allows nighttime. AOD retrievals based on the lunar photometry, in anticipation of a traceable update of ROLO and RIMO models.

Comments

RC: 1. The correction methodology described in the paper in based on the assumption of linear behavior of the instrument with respect to the measured irradiance. The authors need to address this in the paper, to avoid any confusion between instrumental and RIMO correction.

AC: Referee comment is right, we assume that the instrument response is linear. This assumption was confirmed by the study of Taylor et al. (2018); hence, a sentence has been added in the new manuscript to indicate these issues:

"*It is important to remark that this AOD retrieval is based on the assumption of linear behaviour of the instrument with respect to the measured irradiance, but this assumption is reasonable as it was observed by Taylor et al. (2018), who found that nonlinearity can be considered negligible for the CE318-T instrument at Moon irradiance levels.*"

Taylor, S., Greenwell, C., and Woolliams, E.: D3: Lunar Photometer Calibration for Lunar Spectral Irradiance Measurements, Tech. rep.,http://calvalportal.ceos.org/documents/10136/703678/Lunar%2BIrradiance%2BD3%2B-%2BCalibration.pdf, 2018.

RC: 2. What is the spectral uncertainty of the correction? Figure 1 should include a panel demonstrating the uncertainty with respect to MPA as well as the relative RCF to a selected MPA

AC: The proposed RCF correction should only be used for the Cimel 318-T wavelengths: 340, 380, 440, 500, 675, 870, 940, 1020 and 1640 nm, but even the use of 340 and 380 nm is not recommended, as indicated in the conclusions. No wavelength interpolation of RCF should be applied to other wavelengths or spectral bands. It is mainly because the nature of RIMO. RIMO calculates the lunar reflectance at 32 wavelengths that are later interpolated to the CE318-T wavelengths. The accuracy of RIMO in the wavelengths within a spectral range defined by two consecutive RIMO wavelengths (of the 32 wavelengths) can be totally different in other spectral range.

Following the reviewer comment we have calculated the uncertainty on the a, b and c coefficients for the RCF calculation. Figure 1c panel has been modified including the uncertainty of the RCF but also Figure 1 has a new panel (Figure 1d) in the new manuscript version with the spectral variation of RCF for different MPA values. Figure 1c and 1d are shown in Figure R1. The next sentences have been added to discuss the obtained results.

*"The uncertainty on RCF caused by the uncertainty on the coefficients is also shown in Figure 1c. This uncertainty increases with MPA and is in general low except for the UV channels. Figure 1d shows the RCF values as a function of the nominal wavelengths of the photometer channels and for a set of MPA values. The uncertainty of the RCF increases with MPA as observed in Figure 1c. About the variation of RCF with wavelength, it is similar for the different MPA values, being always larger for negative MPA values than for positive ones, except for 1020 channel. The RCF strongly decreases from 340 to 440 nm, while from 440 to 935 nm the variation is smoother, increasing from 440 to 675 nm and decreasing from 675 to 935 nm. This result could lead us to think that RCF can be calculated for other wavelengths by interpolation. However, the spectral variation of RCF is unknown and smooth or linear behaviour cannot be assumed. RIMO lunar reflectance values are calculated at 32 spectral bands which are interpolated to the other wavelengths, the accuracy of RIMO could drastically vary between two different RIMO bands. Therefore, the interpolation of RCF to other bands is not recommended or at least must be taken with care. The spectral uncertainty and accuracy of RCF is not known out of the CE318-T spectral bands."*

[Figure]

Figure R1: a) Fitted RCF and ± its propagated uncertainty vs. MPA for different wavelengths (340 nm values are not shown because they are out of the axis limits). b) Fitted RCF and ± its propagated uncertainty (error bars) against the nominal wavelength of each CE318-T channel, for different MPA values.

RC: 3. Has the RCF been applied to other photometers/spectroradiometers?
AC: The RCF has been applied to other Cimel CE318-T photometers and it works good even at different locations (see González et al., 2020). But this correction has not been applied to other photometer models or spectroradiometers yet. To test how much dependent on the instrument the proposed correction is, we encourage other researchers to validate this method with other instruments. However, we know the RCF was developed only for the CE318-T spectral bands and therefore the extrapolation of RCF to other spectral wavelengths is not recommended. For other instruments the full methodology should be applied in order to retrieve new RCF for their specific spectral bands.
The last sentence of the paper has been modified as follows:

"*Moreover, additional studies using different Moon photometer/spectroradiometer models or using alternative and independent night-time instrumentation, like lidar or star photometers, are highly recommended to characterize the AOD uncertainty, the accuracy of the proposed method and the feasibility of its use with other instrumentation.*"

RC: 4. How the degradation of the reference Cimel is accounted for? Are the daytime calibrations used between the night observation?
AC: The calibration coefficient of each channel is time interpolated between the previous (pre-) and later (post-) AERONET standard calibration for daytime (solar observations for AOD). This interpolated coefficient is transferred to the night-time calibration by the Gain method. Hence the degradation of the instrument is considered by the temporal interpolation between the pre- and post-deployment calibrations.

RC: 5. The stability of the atmospheric aerosol load has been well described, however what is maximum difference between the afternoon and next morning AOD to retrieve the correction factor? Is there any dependency of the RCF to the slope of the linear fit?

AC: There is no threshold for the maximum AOD difference between afternoon and next morning, but for stable and pristine selection we are demanding that the AOD at 500 nm must be below 0.025. Hence, indirectly there are a maximum difference between the afternoon and next morning AOD at 500 nm of 0.025. The dependence of RCF on the slope (AOD variation rate during nights) has not be studied. However previous studies about Izaña (Toledano et al. 2018) indicate no systematic diurnal cycle of the aerosol at the site. Therefore, we are confident about the absence of systematic effects that could bias our results.

RC: 6. Apart from the comparison of the corrected AOD to the star photometer it would be very interesting to add in figures 2,3,4 the uncorrected AOD retrievals, so the reader can visualize the improvement.

AC: The same analysis for the uncorrected AOD has been done. The main problem is the cloud-screening application, since these uncorrected AOD results in Angström Exponent values out of the cloud-screening limits and a lot of cloud-free measurements are rejected. Anyway, the same Moon-Star comparison with uncorrected AOD has been done choosing the data labelled as cloud-free by the RCF-corrected AOD. The next figures (R2, R3, R4 and R5) show the results. It is true that the reader can see how the uncorrected AOD fits worse, underestimating the star AOD (around -0.05) which is more evident as MPA increases. However, the reader knows that because the AOD differences regarding a reference AOD are shown in Figure 1a at Izaña. Moreover, the addition of the uncorrected data to the panels makes them more confusing due to the high number of data points and information. We know that it is important to remark the improvement, but in this case, we assume that the uncorrected data is not useful for AOD calculation and hence we prefer to focus the comparison on the analysis of the proposed method.

[Figure]

Figure R2: Aerosol optical depth (AOD) values from Sun, Moon (with and without RCF correction) and star photometer at Granada (Spain) from the first to third Moon quarter in July 2016. Bottom panel at right shows the Angström Exponent (AE) calculated with the wavelengths of 440, 500 and 675 and 870 nm (436, 500, 670 and 880 nm for star photometer). Moon phase angle (MPA) is represented with a black line in each panel.

[Figure]

Figure R3: Aerosol optical depth (AOD) and Angström Exponent (AE) from Moon photometer with and without RCF correction versus the AOD from star photometer for 2016-2017 period and for different wavelengths. Linear fits are also represented for each wavelength.

[Figure]

Figure R4: Aerosol optical depth (AOD) differences between the Moon and star photometers as a function of Moon phase angle (MPA) for different wavelengths. Bottom-right panel shows these differences for Angström Exponent (AE) in the 440-870 nm range. Black circles represent the median of all differences in a ±5° MPA interval, while error bars indicate ± standard deviation of the data in the same interval.

[Figure]

Figure R5: Frequency of the aerosol optical depth (AOD) differences between the Moon (without RCF correction) and star photometers for different wavelengths. Bottom-right panel shows the frequency of these differences for the Angström Exponent (AE) in the 440-870 nm range.

RC: 7. A spectral RCF version of the Figure 1c for selected MPA would be helpful.
AC: See the answer to the second referee comment.

RC: 8. Why the could-flagging is wavelength dependent? Given the noise of 380 nm why the cloud flag from next measured wavelength is not used?

AC: The development of a robust cloud-screening for AOD at night-time is out of the scope of this paper, as it is explained in the manuscript. As a first step, we translated the cloud-screening for daytime AOD (based on AERONET criteria) to the night-time.

The used cloud-screening is not wavelength dependent. When the algorithm detects clouds then all the wavelengths are removed for the particular observation. The cloud-screening main criteria employ temporal variation thresholds at different time scales, using the infrared channels and 500nm. The low signal at 380 nm can result in bad AOD data in this channel even if the sky conditions (as indicated by the other wavelengths) were cloud free.

We may in the future introduce additional quality-assurance criteria within the screening algorithm, in order to reject channels without realistic AOD data even in the absence of clouds (for instance due to noisy signal or defective filter, etc).

Technical comments/suggestions

RC: Line 2: that is very relevant in polar areas Important, interesting, high value

AC: "Relevant" has been replaced by "important".

RC: Line 14: that provides the expected AOD values provides AOD closer to the expected values

AC: Done.

RC: Line 87: located below the Izaña's level. located below Izaña's level /altitude

AC: "level" has been replaced by "altitude".

RC: Line 121: same detectors as the Sun

AC: The sentence has been changed by:

"*Sky radiance at solar aureole and direct Moon irradiance are measured with the same detectors used to measure direct solar irradiance, but with an electronic amplification factor (gain) of 128 and 4096, respectively.*"

Line 125: the photometers used in this paper belong to AERONET, being the #933 a reference photometer used at Izaña data Used for Izaña data / operated at Izaña What is the measurement period?

AC: "used" has been replaced by "operated". The period is not added in this part since it is always the same photometer used for Izaña data, while for UGR we need to discern between three different photometers which is important in the Moon-Star comparison because some differences could be caused by the photometer. The measurement period chosen for the #933 photometer at Izaña is mentioned in Section 3.3: from June 2014 to March 2018.

RC: Line 165: makes that the knowledge of the absolute extraterrestrial irradiance is not needed in the AOD calculation, because an equivalent Noncompulsory

AC: we have reformulated the paragraph:

"*A main advantage of Sun photometry is that the measured irradiance is directly emitted by the Sun and then, the solar irradiance reaching the top of atmosphere (extraterrestrial irradiance) does not significantly change, at least along one day. The Earth-Sun distance is the main factor modulating this irradiance, causing variations about ±3% along the year. Following the Beer-Bouguer-Lambert law, the extraterrestrial signal of the instrument (rather than irradiance in physical units) is needed for AOD calculation. This can be obtained by the Langley plot method (Shaw, 1976, 1983), in which direct Sun irradiance is observed at different solar elevations in order to extrapolate the top-of-the-atmosphere signal of the instrument. Side by side comparison with a reference instrument is the common practice in AERONET for calibration transfer in field instruments (Holben et al., 1998; Toledano et al., 2018; Giles et al., 2019; González et al., 2020).*"

RC: Line 167: calibration transfer

AC: Done.

Line 170: this fact points out the need of knowledge of the extraterrestrial lunar irradiance for Moon photometry purposes this fact points out the need of knowledge of the extraterrestrial lunar irradiance, and especially the variation with respect to the MPA, for Langley based Moon photometry purposes

AC: the text has been changed as:

"*However, the Moon is not a self-illuminating body. It reflects solar radiation with exceptional stability (Kieffer and Stone, 2005). Due to the changing positioning of Sun, Moon and Earth, lunar irradiance at the top of the Earth's atmosphere significantly changes with the Moon Phase Angle (MPA), even along one single night. This fact points out the need of accurate knowledge of the extraterrestrial lunar irradiance for Moon photometry purposes. In this framework, AOD from lunar irradiance observations can be calculated following the Beer-Bouguer-Lambert law, as follows (Barreto et al., 2013)*"

RC: Line 360: appreciated in Figure 3 since they are out of axis limits, and they are not cloud-screened since the used criteria does not reject Seen

AC: This sentence has been changed to:

"*These values are not cloud-screened because the removal of negative AOD values is not included in the screening algorithm. These negative values are the main cause of the shifted linear fit shown in Figure 3 for 380 nm. This plot, however, shows that there are many data points of AOD (380 nnm) close to the 1:1 line.*"